# Nanoforest: Polyaniline Nanotubes Modified with Carbon Nano-Onions as a Nanocomposite Material for Easy-to-Miniaturize High-Performance Solid-State Supercapacitors

**DOI:** 10.3390/polym10121408

**Published:** 2018-12-19

**Authors:** Piotr Olejnik, Marianna Gniadek, Luis Echegoyen, Marta E. Plonska-Brzezinska

**Affiliations:** 1Institute of Chemistry, University of Bialystok, Ciolkowskiego 1K, 15-245 Bialystok, Poland; 2Department of Chemistry, University of Warsaw, Pasteur 1, 02-093 Warsaw, Poland; mgniadek@chem.uw.edu.pl; 3Department of Chemistry, University of Texas at El Paso, 500 W. University Ave., El Paso, TX 79968, USA; echegoyen@utep.edu; 4Faculty of Pharmacy with the Division of Laboratory Medicine, Medical University of Bialystok, Mickiewicza 2D, 15-222 Bialystok, Poland

**Keywords:** polyaniline nanotube, carbon nano-onion, conducting polymer, nanocomposite

## Abstract

This article describes a facile low-cost synthesis of polyaniline nanotube (PANI_NT_)–carbon nano-onion (CNO) composites for solid-state supercapacitors. Scanning electron microscopic (SEM) analyses indicate a uniform and ordered composition for the conducting polymer nanotubes immobilized on a thin gold film. The obtained nanocomposites exhibit a brush-like architecture with a specific capacitance of 946 F g^−1^ at a scan rate of 1 mV s^−1^. In addition, the nanocomposites offer high conductivity and a porous and well-developed surface area. The PANI_NT_–CNO nanocomposites were tested as electrodes with high potential and long-term stability for use in easy-to-miniaturize high-performance supercapacitor devices.

## 1. Introduction

Electronic technology has been intensively developed over the last several decades. New research trends are focused on creating novel, fast-responding, and miniaturized electronic devices. To increase the energy, semiconductors have been replaced by carbon and organic materials such as proteins, conducting polymers, or their combinations. Such combinations of at least two materials of different chemical nature are called composites [1,2,3]. Previous research investigated polymers containing a π–electron conjugated system in their structures, e.g., polyaniline (PANI), polythiophene, and polypyrrole. These polymers are characterized by high values of the specific conductivity, which can be controlled by the oxidation state, pH [4], and type of dopant ions [5]. Such flexibility, in combination with their properties, corrosion resistivity, and chemical neutrality opens up numerous possibilities for their application in electronic devices.

PANI is a pioneering representative of the conducting polymer group. The PANI chains consist of −p-coupled aniline units (Scheme 1) [6]. The combination of benzenoid and quinoid rings leads to different oxidation states for the PANI polymer: leucoemeraldine, emeraldine, and pernigraniline. Due to the presence of negative polarons, the green emeraldine form of PANI exhibits electroconductive properties. In addition to advantages such as a high conductivity of 9 S cm^−1^ [7], high chemical stability, and low-cost chemical or electrochemical preparation methods, this polymer also exhibits potential capacitive properties. The capacitance strongly depends on the chemical and physical properties of the polymer, which are frequently a consequence of the synthesis procedure.

The effectiveness of supercapacitors (SCs) is determined by several important material factors, including electroconductivity, type of dopant ions, specific surface area, morphology, and pore size, as well as the material arrangement, distribution, and orientation with respect to the surface [8]. In particular, morphology is a crucial parameter for solid-state devices, because it can increase the interface between an electrode and electrolyte [9]. Therefore, nanostructured conducting polymers have received attention due to their high surface area-to-volume ratio and high surface free energy [10]. There are several synthetic pathways for aniline oxidation and different types of nanostructure production, which lead to the formation of different phases for PANI. The most popular methods are chemical oxidation [11], template methods [12], and electrochemical processes [13], with less popular methods involving sonochemical [14] or radiation approaches [15]. Dhawale et al. have reported a specific capacity of 503 F g^−1^ for bulk PANI synthesized via a chemical bath deposition method measured at a sweep rate 10 mV s^−1^ in 1 M of H_2_SO_4_ [16]. For comparison, the specific capacity for a PANI nanofiber-modified electrode obtained in the same acid solution in the presence of an ammonium persulfate oxidant, was determined to be 235 F g^−1^ [17]. The above-mentioned methods enable one to realize randomly aggregated granules, nanoroughened polymer hydrophobic surfaces, nanospheres surrounded by surfactant molecules, nanofibers, or nanotubes inside the membrane matrix [18]. 

One of the most common techniques for the formation of polymer nanostructures is template synthesis, which is often used in the controlled fabrication of PANI nanotubes (PANI_NT_) [12,19]. Template selection enables control over the nanotube length and its internal cavity diameter, which consequently affects the nanostructure’s properties. The mechanism for nanotube formation is based on the aniline nucleating a stacking process, which is stabilized by π–π interactions between phenazine structures [20]. Due to the strong intermolecular interaction between polymer chains, PANI_NT_ reveals high conductivity relative to bulk polymer films [21,22,23]. Martin et al. confirmed experimentally that the conductivity for randomly distributed PANI_NT_ depends on the nanotube size, and is six times higher than that of the macromolecular polymer [7]. Therefore, the electric properties for nanostructures (conductivity and capacity) can be enhanced by increasing the degree of material order [24,25]. In the case of PANI_NT_, the most effective ordering occurs for nanostructures that are oriented perpendicular to the surface. Such arrangement enables easier internal and external nanotube modification, which can significantly increase the total specific surface area, which is the most important parameter in the SC field [26,27]. PANI_NT_ are also only slightly soluble in common organic solvents, which is crucial for chemical stability.

Despite the many advantages of conducting polymers, they also exhibit certain disadvantages that limit the polymers’ practical use in SCs. For example, these polymers cannot be utilized on their own as SC electrodes due to a poor power density, low charge exchange rates, and poor long-term stability during the charge–discharge processes, which lead to electrode damage [28]. To overcome these disadvantages, composite materials are frequently used as electrodes for SCs. Material systems often used as electrodes in SCs are carbon nanomaterials (CNs) and conducting polymers. Combining conducting polymers with CNs mainly enhances the specific surface area, inducing high porosity, facilitating electron and proton conduction, increasing the number of active sites, protecting active materials from mechanical degradation, and improving cycling stability [29,30,31,32]. PANI composites with CNs, such as single-walled carbon nanotubes (SWCNTs), multi-walled carbon nanotubes (MWCNTs) or graphene (G) and graphene oxide (GO) sheets, have been reported. The specific capacitance for the above listed PANI/carbon nanocomposites are: 485 F g^−1^, 560 F g^−1^, 413 F g^−1^, and 375 F g^−1^, respectively [33,34,35,36]. 

In this report, we focus on the synthesis of a composite containing PANI_NT_ and multilayered fullerenes, frequently called as carbon nano-onions (CNOs). CNOs consist of a hollow spherical fullerene core surrounded by concentric and curved graphene layers with progressively increasing diameters. The interlayer distance between neighboring layers is 0.335 nm [37,38]. These CNOs can have different sizes and shapes, which, in turn, determine their physical properties and chemical reactivity [39,40,41,42]. In our study, we used small spherical CNOs obtained by the graphitization of nanodiamond particles (NDs, 5 nm) at high temperature under partial vacuum [42,43]. These CNOs show the unique combination of mechanical properties with chemical and physical properties [44,45]. They possess a relatively high surface-to-volume ratio, high conductivity, and high thermal stability. These properties, with the combination of satisfactory compatibility, can lead to the preparation of composite materials. Their high reactivity when compared with CNs enables one to create homogeneous three-dimensional (3D) composite materials using both organic and inorganic components [45,46,47,48]. The reactivity of fullerene-like structures, including CNOs, decreases with increasing size due to a decrease in the curvature of the surface, due to decreased strain. Additionally, the ability to functionalize CNO surfaces depends on the presence of defects on the carbon surface as well as on the presence of carbon atoms with *sp*^2^ hybridization. The integration of CNOs with other substances can lead to interesting materials possessing properties of the individual components. In particular, the combination of CNOs with conducting polymers yields new materials [29,30,31,32], which are highly attractive as electrode materials for electrochemical and biomedical purposes. We have already emphasized that the organization of the two components in the matrix, and the scale on which this occurs, have a decisive influence on the physicochemical properties of the synthesized materials. 

## 2. Materials and Methods

### 2.1. Materials

Aniline monomer and sulfuric acid 95–97% were purchased from POCh (Gliwice, Poland). Ammonium persulfate 98% (NH_4_)_2_S_2_O_8_, *N*-hydroxysuccinimide (NHS), and 1-ethyl-3-(3-dimethyl aminopropyl) carbodiimide hydrochloride (EDC) were purchased from Sigma Aldrich (Saint Louis, Missouri, USA) and used as received. Chloroform was obtained from Chempur (Piekary Slaskie, Poland). Aluminum oxide powder was purchased from Buehler Micropolish (Esslingen, Germany). All of the reagents (p.a. grade) were used without further purification. All of the solutions were prepared using water purified by a Milli-Q system from Merck (Darmstadt, Germany) with a resistivity of 18.2 MΩ and pH of 7. 

### 2.2. Polyaniline Nanotube Matrix Synthesis

PANI_NT_ synthesis was accomplished by the template method. Whatman Nuclepore polycarbonate (PC) membranes (with diameter 200 nm) were used as templates. The synthesis process was conducted in 1 M of sulfuric acid medium by the chemical oxidation of the aniline monomer using ammonium persulfate as oxidant. In a typical experiment, a PC membrane was coated on one side with a thin, uniform gold film using a plasma sputter coater (Leica ACE 200, Wetzlar, Germany) by vapor deposition. The modified template was then soaked in five mL of 0.3 M of aniline acidic medium for 30 min before being mixed with the same volume of 0.3 M of (NH_4_)_2_S_2_O_8_ in one M of sulfuric acid solution. The reaction vessel was kept at a low temperature (~4 °C). The typical reaction time was approximately three hours. Subsequently, the membrane was dissolved in chloroform and removed. Next, the separated thin gold film with PANI_NT_ was carefully rinsed with deionized water.

### 2.3. Synthesis of Pristine and Oxidized CNOs 

Pristine CNOs: Commercially available nanodiamond powder (NDs, Carbodeon μDiamond^®^Molto, Vantaa, Finland) with a crystal size between four and six nm and nanodiamond content larger than 97 wt %), was used for the preparation of spherical CNOs using the procedure proposed by Kuznetsov et al. [49] NDs were placed in a graphite crucible and transferred to an Astro carbonization furnace. Annealing of the ultradispersed NDs was carried out at 1650 °C under a 1.1 MPa He atmosphere using a heating rate of 20 °C min^−1^. The final temperature was maintained for one hour; then, the material was slowly cooled to room temperature. The furnace was opened, and the CNOs were annealed in air at 400 °C to remove any amorphous carbon.

Oxidized CNOs (CNOs_ox_): The oxidation of pristine CNOs was conducted as originally described by Lieber et al. for SWNT [50], and later applied to CNOs in our laboratory. Then, 100 mg of pristine CNOs was dispersed by ultrasonication for 30 min and refluxed for 48 h in 3.0 M of aqueous nitric acid. The mixture was later centrifuged for 10 min followed by collection of the black powder that formed in the bottom of the test tube. Then, Salzmann’s protocol was applied to purify the oxidized CNOs (CNOs_ox_) [51]. The resulting oxidized product was stirred in 3.0 M of NaOH and washed several times with distilled water until a final pH of 7 was reached, and then dried overnight at 110 °C.

### 2.4. Methods

The PANI nanotubes/CNOs_ox_ layers deposited on the electrode surface were studied using a FEI Tecnai S-3000N (Tokyo, Japan) and a Merlin (Zeiss, Germany) field-emission scanning electron microscope (SEM). The CNOs_ox_ nanostructures were examined by a transmission electron microscope (TEM) system Libra 120 (Zeiss, Germany). A digital optical microscope HIROX KH-87000 (Tokyo, Japan) was used for the preliminary observation of the nanocomposite material morphology and arrangement. 

The infrared spectra were recorded using a NICOLET IN10 MX infrared microscope (Thermo Scientific, Waltham, Massachusetts, USA). The microscope was operated mainly in reflectance mode, and the Mercury-Cadmium-Telluride (MCT) detector cooled with liquid nitrogen. The spectra were collected for a 100-µm (area 0.01 mm^2^) square region of the sample. For typical measurements, the spectral resolution was 4 cm^−1^, and 256 scans were averaged to obtain a single spectrum. The spectrum of the pristine CNOs_ox_ was recorded in a potassium bromide (KBr) pellet using the microscope in transmission mode. Additionally, the above-mentioned MCT detector was utilized for mapping the nanostructural layers. 

The Raman experiments were carried out using a Renishaw Raman InVia Microscope (Wotton-under-Edge, United Kingdom) equipped with a high-sensitivity ultralow-noise Charge Coupled Device (CCD) detector. The Raman module was equipped with a microstage that enabled the measurement of a sample in reflectance mode. The instrument was operated using an Ar ion laser with the 514-nm excitation line. For typical measurements, the spectral resolution was 4 cm^−1^, with three scans (each of 10-s duration) averaged to obtain a single spectrum. 

The electrochemical experiments were carried out using an AUTOLAB (Utrecht, The Netherlands) potentiostat/galvanostat with the NOVA software from AUTOLAB (Utrecht, The Netherlands). A typical three-electrode configuration was used with a glassy carbon (GC) disk electrode (two mm diameter) as the working electrode, Ag/AgCl (with saturated KCl) as the reference electrode, and a platinum mesh as the auxiliary electrode. The geometrical area of the glassy carbon electrode was equal to 0.0314 cm^2^. The working electrode was polished with 0.5-µm alumina powder on a polishing wheel, and subsequently washed thoroughly several times with deionized water and ethanol, before being allowed to dry at room temperature. All of the measurements were performed in anaerobic conditions at room temperature (22 ± 2 °C). To remove all of the dissolved oxygen, the measuring cell was Ar-purged 15 min before the experiments began.

## 3. Results and Discussion

### 3.1. Nanocomposite PANI_NT_/CNOs_ox_ Electrode Preparation Procedure

Scheme 2 and Scheme 3 show the simplified procedures that were used for the covalent functionalization of PANI_NT_ with CNO_ox_, which resulted in the creation of the nanocomposites. Briefly, the composite preparation procedure was based on two steps. In the first approach, PANI_NT_ synthesis was accomplished by the template method described in detail in the Experimental section and schematically presented in Scheme 2. After removal of the PC membrane, the organized PANI_NT_ layer was formed on a gold surface.

Next, the GC electrode was repeatedly covered with an Au/PANI_NT_ nanotube film. The covalent functionalization of the Au/PANI_NT_ layers with CNOs_ox_ was promoted via water-soluble carbodiimide (EDC) and *N*-hydroxysuccinimide (NHS). This step was carried out without contact with PANI_NT_. The procedure used was as follows: initially, one mg of CNOs_ox_ was placed in a solution of 10 mM NHS and 40 mM EDC for one hour (Scheme 3). During this reaction, the carboxylic groups of the CNOs were transformed into reactive *N*-hydroxysuccinimide esters. After the activation step, CNOs_ox_ without solvent were added to one mL of ethanol, and the mixture was ultrasonicated for 0.5 h to obtain a dusky gray, uniform, and stable suspension. In the final step, the activated CNO_ox_ suspension was transferred to the Au/PANI_NT_ surface, and after the formation of the amide bonds, the excess of unreacted carbon nanoparticles in the solution was removed from the electrode surface. The formed Au/PANI_NT_/CNO_ox_ layers were tested as supercapacitors.

### 3.2. Raman and Infrared Spectroscopy Studies of PANI_NT_/CNOs_ox_

Raman and infrared spectroscopy were utilized as the main experimental techniques for the qualitative characterization of the composite materials containing the carbon nanoparticles. Figure 1 shows the Raman spectrum of the oxidized CNOs. The spectrum was excited at a wavelength of 514 nm. In general, the spectrum is composed of four characteristic peaks [52], which correspond to the contribution of the hexagonal mode characteristics of graphene or graphite. The most distinctive signal at approximately 1577 cm^−1^ is called the *G* band, which corresponds to the in-plane optical mode of vibration for two adjacent *sp*^2^ carbon atoms on an ideal hexagonal ring of graphite. The *G* bandwidth depends on the amount of deformed chains and hexagonal rings. A wider *G* band corresponds to a lower order in the structure [53]. The spectra are dominated by the *D* band at 1340 cm^−1^. The presence of the *D* band is due to defects in the carbon crystalline curved structure. The larger *D* band intensity is connected with a higher structural disorder, which is caused by the presence of oxygen functional groups on the CNOs_ox_ surface. Additional combined tones for the peaks are located at 2674 cm^−1^ (*2D*) and 2925 cm^−1^ (*D* + *G*). The *2D* band reflects a two-photon process engaging phonons with opposite wave vectors. 

Figure 2 shows the typical Raman spectra for the vertically oriented PANI_NT_ and the PANI_NT_/CNOs_ox_ nanocomposite. The spectra were also excited at a wavelength of 514 nm using a He–Ne laser. The applied excitation frequency falls in the absorption range of PANI_NT_, thereby affecting the spectral enhancement, which is slightly shifted relative to that observed for the macromolecular form of PANI [54]. The low wavenumber region for the pristine PANI_NT_ spectrum contains bands at 520 cm^−1^ and 814 cm^−1^ corresponding to N–H and C–H out of plane deforming vibrations of the quinonoid ring, respectively (Figure 2B). The signal at 573 cm^−1^ is assigned to phenoxazine and phenazine-type unit vibration [55,56]. The band near 1170 cm^−1^ is attributed to the C–H bending vibrations for the bipolaronic, semi-quinonoid rings. This band includes a less visible shoulder at 1192 cm^−1^ connected with C–H in-plane benzenoid ring bending. The signals at 1335 cm^−1^ and those near 1250 cm^−1^ are characteristic for charge carriers and correspond to delocalized polaronic units and ring deformation vibrations, respectively [57,58]. This indicates that the polymer nanotubes are in a conductive form. The PANI_NT_ spectrum also exhibits two specific peaks: a single peak at 1496 cm^−1^, which is connected to the C=N stretching mode of the quinonoid units, and double peaks in the range of 1518 cm^−1^ to 1620 cm^−1^, which provide information for the C–C and C=C stretching vibrations in the above-mentioned structures [59]. In the case of the PANI_NT_/CNOs_ox_ nanocomposite (Figure 2A), the spectrum confirms the presence of carbon and polyaniline nanostructures, and contains previously described characteristic signals. 

The PANI_NT_/CNOs_ox_ composite was also characterized by Fourier transform infrared (FTIR), as shown in Figure 3. The nanocomposite spectrum (Figure 3A) does not differ much from that for the pristine polymer nanotubes (Figure 3B). The most typical signals are located at 837 cm^−1^ and 1165 cm^−1^, which correspond to the C–H out-of-plane deformations and in-plane bendings in the benzene ring [60]. The peaks assigned to 1504 cm^−1^ and 1589 cm^−1^ are connected with the characteristic C=C stretching vibration of the benzenoid and quinonoid rings, respectively. The bands near 1225 cm^−1^ and 1310 cm^−1^ originate from the C=N and C–N stretching vibrations, respectively. The broad signals at higher frequencies (3000–3500 cm^−1^) are connected with the free N–H stretching vibrations [61,62]. The presence of CNOs_ox_ in the composite structure is confirmed by the poorly defined peak at 1760 cm^−1^, which can be assigned to the carbonyl group stretching vibrations. The oxidized CNOs beyond carbonyl groups may also contain different surface functional species, including oxygen (Figure 3C) [63]. The increased intensity in the high frequency range (2900–3500 cm^−1^) could also indicate the successful functionalization of the CNO surface by hydroxyl groups.

Figure 4 shows measurements performed using an infrared mapping method. The measurement maps show the optical distribution of specific signals, with the signal intensity imaged using an appropriate color. The red color indicates the highest intensity signal or the whole spectrum. The images captured by an optical microscope reveal a large fragment of the PANI_NT_/CNOs_ox_/Au surface with dimensions of 700 µm × 800 µm. A point spectrum for the nanocomposite material containing all the characteristic signals described and present in Figure 3A is shown in Figure 4D. Figure 4A illustrates the distribution profile for the PANI_NT_/CNOs_ox_ specific spectrum, and indicates the total surface coverage with a uniform nanocomposite film. A high degree of surface coverage is one of the most important parameters for electrode construction and good performance. The arrangement profile for the conductive PANI_NT_ (Figure 4B) based on the characteristic peak (1589 cm^−1^) shows the presence of nanotubes across the entire experimental area. In the case of the CNOs_ox_, the infrared map shows more blue areas, which indicates a lower intensity for the C=O stretching vibrations for the surface oxygen functional groups (Figure 4C). This finding reflects the small size of the CNOs and their easy aggregation.

### 3.3. Nanocomposite Morphology Study

The morphology of the PANI_NT_/CNOs_ox_ nanocomposite and individual nanostructural components such as pure polyaniline nanotubes and pristine CNOs was characterized using field-emission scanning electron microscopy (SEM), transmission electron microscopy (TEM), and optical microscopy at light field mode (Figure 5). The TEM image for pristine CNOs randomly dispersed on a copper mesh is shown in Figure 5A. The TEM image clearly reveals visible CNO nanoparticles with spherical structures with an average diameter of five nm. The CNO structures exhibit concentric graphitic layers. The number of graphene walls in one carbon nanoparticle varies between six and ten. The TEM diffraction patterns also indicate that the distance between individual spheres equals 0.33 nm, which corresponds to the dimension in pyrolytic graphite [38]. The size of the CNO and its strain makes them an ideal nanoparticle for further functionalization and incorporation into larger systems, despite their strong predisposition to aggregation, which is also shown in Figure 5A. To minimize aggregation, an oxidation reaction was applied, which successfully increases the hydrophilicity of the carbon nanoparticles and increases their dispersibility in polar solvents.

The synthesis of conductive PANI, which was used to form nanotube structures, was accomplished by the template procedure (Scheme 2). The smooth flat surface of the PC membrane, which was used as a template, is also shown in Figure 5B. The diameter of the pore sharply defines the diameter of the polymer nanotubes, which in this case is equal to 200 nm. The density of the pore distribution in the PC membrane reflects the amount of the PANI_NT_ structures formed on the surface. Additionally, the one µm of PC membrane thickness determines the length of the PANI_NT_ structures. Therefore, the membrane that is used defines the size of the nanotubes in three dimensions. Figure 5C confirmed that the nanotubes formed during the polymerization process are unfilled and empty inside. The average diameter of the polymer nanotubes is 200 ± 30 nm, and depends on the side-wall thickness. The PANI_NT_ are not completely straight due to the membrane removal via repeatable steps, but they do not show cavities, and are free of solvent and melted PC. The PANI_NT_ were obtained as a randomly assembled nanostructure (Figure 5D,E), where chain aggregates are formed that rise vertically to the surface (Figure 5F–H). The second step for the synthesis required a prior sputtering of a thin gold layer onto the membrane. The sputtered 100-nm thick gold films completely blocked the pores on one side, and became a substrate for the growth of PANI_NT_. Figure 5F,G indicates that the polymer nanotubes extend perpendicularly to the surface, creating a brush-like “nanoforest”. The PANI_NT_ orientation provides a larger active surface area for the conductive polymer, which enables greater availability, resulting in better efficiency for the further functionalization with CNOs_ox_. There is also a higher probability of filling the empty core of the nanotubes by carbon nanoparticles, which have a diameter that is approximately 20 times smaller. An active and highly developed surface area is the most important parameter for materials that are used as electrodes in supercapacitor devices, which defines their electrochemical properties. The capacitance of such systems is directly proportional to the surface of the electrodes that is available for transport of the electrolyte ions. 

Figure 6 shows SEM images for PANI_NTs_ modified with CNOs_ox_ at varying concentrations. The functionalization of the polymer nanotubes with CNOs_ox_ was carried out in the presence of water-soluble EDC and NHS, as described previously (Scheme 3). Despite the very small size of the carbon nanoparticles (~5 nm), the SEM images do not exhibit single and separated CNOs_ox_ particles. The van der Waals forces between the oxidized carbon nanoparticles lead to self-aggregation and the formation of nanoclusters, and their amount and distribution are concentration-dependent (Figure 6B–D). The aggregates of carbon nanoparticles with different dimensions formed a spongy-like structure. The CNO_ox_ particles were accumulated both between and onto PANI_NT_. The SEM images show the difference between pristine PANI_NT_ (Figure 6A) and PANI_NT_/CNO_ox_ nanocomposites, even for low concentrations of CNOs_ox_ (Figure 6B).

### 3.4. Voltammetric Studies of the PANI_NT_/CNOs_ox_ Nanocomposite

The PANI_NT_/CNO_ox_ nanocomposites and undoped PANI_NT_ as GC/Au-PANI_NT_/CNO_ox_ and GC/Au-PANI_NT_ were examined using cyclic voltammetry (CV). Nanocomposites anchored to a thin gold film were immobilized onto the GC electrode surface (Scheme 2). Such a system enables the evaluation of the electrochemical performance and charge storage ability of these systems. The measurements were conducted in one M of sulfuric acid solution within the 0–0.8 V potential range versus Ag/AgCl. The voltammetric curves were recorded using different sweep rates of up to 100 mV s^−1^. The voltammetric responses for pure PANI_NT_ and PANI_NT_/CNO_ox_ (four mg mL^−1^ of CNOs) composites are shown in Figure 7. The PANI_NT_ film exhibited good mechanical and electrochemical stability under cyclic voltammetric conditions within the applied potential range (Figure 7A). Figure 7A,B present the 10^th^ cycle of the CV measurements, and the shape of the CV curves remain essentially unchanged. The characteristic CV response for pristine PANI_NT_ in acidic medium consists of two pairs of redox couples (A_1_/C_1_ and A_2_/C_2_) corresponding to two-electron processes. The peaks A_1_/C_1_ within the 0–0.25 V potential range are attributed to the electrochemical transition between semiconducting leucoemeraldine and the conductive emeraldine form. The peaks A_2_/C_2_ occurring in a more positive potential range are attributed to the benzoquinone to aminoquinone transformation [64].

The PANI_NT_/CNOs_ox_ film exhibited stable and conductive behavior under cyclic voltammetric conditions within this potential range (Figure 7B). The capacitance current depends on the sweep rates and the film composition. The conductivity of this composite arises mainly from the CNOs_ox_ component. The electrochemical responses also indicate the presence of a pair of redox peaks, confirming the contribution of the PANI nanostructures to the capacitance of the composites. The less clarity of PANI_NT_ redox peaks even at low scan rates is caused by the CNOs_ox_ presence, which restricts the electrolyte access to the polymer nanotubes. The PANI_NT_ signals decreased with the increasing sweep rate. The voltammograms for the PANI_NT_/CNOs_ox_ measured at sweep rates higher than 50 mV s^−1^ show almost pseudo-rectangular anodic and cathodic profiles, which reflects a practically ideal double-layer capacitance behavior. The capacitive current varies linearly with the sweep rate below 50 mV s^−1^ at +0.3 V versus Ag/AgCl, as shown in Figure 7C. The deviations from linear dependence of the capacitive current above 50 mV s^−1^ are the results of the electrolyte diffusion limitation. The capacitive current (*I_c_*) is given by Equation 1:*I_c_* = *C_s_ v m*(1)
in which *C_s_* is the specific capacitance, *m* is the mass deposited onto the electrode surface, and *v* is the potential sweep rate. It should be noticed that the mass parameter is directly connected to the active surface of the material, according to Equation (2):*m* = *A ρ_A_*(2)
where *ρ_A_* is the average area density, and *A* is the active surface area of material. The values for *C_s_* calculated from the dependence of the current on the different sweep rates for undoped PANI and the composites are collected in Table 1. The *C_s_* for the undoped PANI and PANI_NT_/CNOs_ox_ composite using CV was also determined from the following Equation (3):(3)$$C_{S}=\frac{\int_{E_{2}}^{E_{1}}i\left(E\right)dE}{vm\left(E_{1}-E_{2}\right)}$$
where *E_1_* and *E_2_* are the initial and final potentials (V), respectively, $\int_{E_{2}}^{E_{1}}i\left(E\right)dE$ is the integrated current over the potential window, *v* is the sweep rate (V s^−1^), and *m* is the mass of the active material. The values of the specific capacitances obtained by the integration of *I* versus *E* curves are slightly different compared to those calculated from the linear relationship for the *I* versus *v* plots (Table 1). A larger difference of the calculated *C_s_* values is observed for low sweep rates (<5 mV s^−1^). For both cases, the specific capacitances for the nanocomposite are higher than those obtained for the pristine PANI_NT_.

The capacitance value calculated using Equation (2) at 1 mV s^−1^ is 946 F g^−1^, which is much higher than that for pristine PANI_NT_ (269 F g^−1^) (Table 1). It is possible that this is due to the PANI_NT_/CNOs_ox_ surface area increase while maintaining the electroactive behavior. The dependence of the specific capacitance versus scan rate shows that the PANI_NT_/CNOs_ox_ nanocomposites are capable of storing more electric charge compared to pristine PANI_NT_, regardless of the sweep rate. It is also important to note that the specific capacitance for PANI_NT_ reveals a more linear behavior compared to that for the nanocomposites within the same sweep rate range. When the sweep rate was increased to 10 mV s^−1^, the capacitive current for the nanocomposite decreased and represented only ca. 65% of the starting value. However, the shape of the CV curves remain essentially unchanged even at high scan rates, suggesting that the electrode exhibits excellent charge transport, while the gravimetric capacitance gradually decreased upon increasing the scan rate.

The data show that the PANI_NT_/CNO_ox_ nanocomposites are ideal materials for supercapacitors. Compared to other systems described in the literature that contain carbon nanoparticles and PANI (Table 2), our nanocomposites exhibit better electrochemical properties, including a notably higher specific capacitance. The higher values of specific capacitance for the PANI_NT_/CNOs_ox_ nanocomposite result from the high conductivity of both nanostructures, due to their extremely high porosity and organized brush-like structures. In particular, “conductive” channels were created in which the interactions between π-electrons of the PANI aromatic/quinonoid structures and CNO graphitic layers facilitate charge transport. The high effectiveness of supercapacitor devices containing PANI_NT_/CNOs_ox_ can also be realized due to the specific nanocomposite architecture, in which the nanotubes are oriented vertically to the surface, thus providing easy access for the electrolyte and facilitating ion diffusion.

Values for the specific capacitance of undoped PANI and composites containing this polymer and other CNs measured at low sweep rates are collected in Table 2. As observed from the CVs, the composites exhibited better electrochemical performance compared to most of the undoped conducting polymers. Additionally, it should be noted that the electrochemical properties of the composites are affected by the type of carbon nanostructures and the form of the conducting polymer.

## 4. Conclusions

We demonstrated that nanocomposites containing PANI nanotubes and carbon nano-onions can be prepared by the template method. The combination of these two types of materials improved the capacitive properties. Notably, the nanostructural properties of both components and the unique perpendicular organization of the conducting nanotubes relative to the surface electrode affected the unusual electrochemical properties of these materials. The electrochemical performance of the composites is affected by the mass of the carbon nanostructures. The PANI_NT_/CNO_ox_ composites exhibited a high specific capacitance ca. 950 F g^−1^, which is one of the highest values published to date for analogous materials. The main advantage of these composites is their potential for use as conductive materials in solid-state supercapacitors.

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
