# Peer review of "Nanoforest: Polyaniline Nanotubes Modified with Carbon Nano-Onions as a Nanocomposite Material for Easy-to-Miniaturize High-Performance Solid-State Supercapacitors"

_polymers, 2018, doi:10.3390/polym10121408_

Round 1
Reviewer 1 Report
The authors demonstrateda supercapacitor electrode containing PANI nanotubes and carbon nano-onions by template method. The work in itself is quite interesting and the authors offer some experiment analysis to explain what they observe. The manuscript addresses an interesting topic with good background information. While the reviewer believes that the manuscript would be of interest to the readers of Polymers, the reviewer also believes that the manuscript needs major revision to address several following key issues.
1. The resolution in some of the figures is low, and cannot see the words inside the figures either. Please also include all the detailed captions for each figure/subfigure so that the readers can quickly grasp the main idea of the manuscript.
2. Authors mentioned figure 6 is the SEMS images at different concentration, what are the concentrations? Cannot really tell the difference between figure 6B and 6C.
3. How did you attach the PANI/NT/Au onto GC electrode? What electrode system did you use for the electrochemical test? (2 or 3, needs to be mentioned in the manuscript).
4. The authors claimed excellent mechanical and electrochemical stability of the PANI/NT from Figure 7A. However, there is no context of which cycle is it, how does it compare to the 1st cycle. Normally, this claim is backed by the cycling test, and show the capacitance vs. cycle number. In addition, what scan rate was used?
5. There is no comparison between pristine PANI and PANI/NT and PANI/NT/CONS. It would be better for the authors to show why it is better to have those nanoparticles incorporated into PANI.
6. Where are the Table 1 and 2? Is the pristine PANI CV done also at 1 mV/s?
7. Have you done Charge-Discharge experiment to reveal the ESR of the supercapacitor electrode? In addition, N2 absorption is a good way to confirm the hypothesis of the surface area increase. It will also reveal the pore size distribution and average pore sizes too.
Author Response
Thank you for your kind reconsideration of this manuscript. Please find the responses to your comments. The main remark regarding the manuscript. We added all the missing signatures to the Figures and Schemes. Additionally, we added two tables that were missing in the final version of the manuscript (please find attached file). Please find below, our answers to all your comments.
We do hope that you will find the article suitable for publication in its present.
Many thanks indeed for your kind help and consideration.
Sincerely yours,
Marta E. Plonska-Brzezinska
Responses to Reviewer #1 comment
Reviewer #1: The resolution in some of the figures is low, and cannot see the words inside the figures either. Please also include all the detailed captions for each figure/subfigure so that the readers can quickly grasp the main idea of the manuscript.
Comment: The resolution and the quality of all Figures and Schemes were increased. The captions for all Figures and Schemes were included and corrected.
Reviewer #1: Authors mentioned figure 6 is the SEMS images at different concentration, what are the concentrations? Cannot really tell the difference between figure 6B and 6C.
Comment: The SEM images, which are presented in Figure 6, show the images for the pristine PANINT and PANINT/CNOsox nanocomposite. In the present form, the captions included information about the concentration of CNOsox : (0.5 mg/mL (B) 1 mg/mL (C) and 4 mg/mL (D)). The difference of the CNOsox concentration between Fig. B and Fig. C may be not clearly visible because it is only 0.5 mg/mL.
Reviewer #1: How did you attach the PANI/NT/Au onto GC electrode? What electrode system did you use for the electrochemical test? (2 or 3, needs to be mentioned in the manuscript).
Comment: The procedure of the nanocomposite PANINTCNOsox electrode preparation was described in Results and discussion in Subsection 3.1. The PANINT was received as nanotubes perpendicularly attached to the thin gold film. Such film as a chloroform suspension was physically adsorbed, spread on the GC electrode surface and left to dry. The electrochemical tests were carried out using GC/Au-PANINT and GC / Au-PANINT / CNOsox systems what is included in subsection 3.4 and in description of Fig. 7.
Reviewer #1: The authors claimed excellent mechanical and electrochemical stability of the PANI/NT from Figure 7A. However, there is no context of which cycle is it, how does it compare to the 1st cycle. Normally, this claim is backed by the cycling test, and show the capacitance vs. cycle number. In addition, what scan rate was used?
Comment: The cyclic voltammograms, Fig. 7, present 10th cycles of the electrochemical test. In the case of GC / Au-PANINT CV presented in Fig. 7A the scan rate is 1 mV/s, the Fig. 7B shows CVs for GC / Au-PANINT / CNOsox system at: 1, 5, 10, 20 and 100 mV/s in 1M H2SO4 – that information is included in Fig. 7 caption.
The presented cycles do not differ from 2nd scan of the electrochemical test. The first cycle shows always some differences, especially for conducting polymers or their composites. The capacitance of studied systems is stable what proves the electrochemical stability of the investigated systems. Some short comments were added on Page 11.
Reviewer #1: There is no comparison between pristine PANI and PANI/NT and PANI/NT/CONS. It would be better for the authors to show why it is better to have those nanoparticles incorporated into PANI.
Comment: The comparison between GC / Au-PANINT and GC / Au-PANINT / CNOsox systems has been included in subsection 3.4. We describe that the CNOsox presence cause less clarity of redox peaks what is connected with restricted access of the electrolyte to the polymer nanotubes. What is more we compare the capacitance values: 946 F/g for PANINT / CNOsox and 269 F/g for pristine PANINT at 1mV/s. The presence of CNOsox definitely increased the capacitance values what proved their unique electrochemical properties and impact on the surface development what is crucial for materials use in supercapacitor.
Reviewer #1: Where are the Table 1 and 2? Is the pristine PANI CV done also at 1 mV/s?
Comment: Table 1 and 2 were included in the manuscript (please find the attached file). For the clarification two Tables are also included below. The Table 1 present CVs for pristine PANINT both at 1 mV/s and 100 mV/s.
Reviewer #1: Have you done Charge-Discharge experiment to reveal the ESR of the supercapacitor electrode? In addition, N2 absorption is a good way to confirm the hypothesis of the surface area increase. It will also reveal the pore size distribution and average pore sizes too.
Comment: The structure and construction of the nanocomposite material Au-PANINT / CNOsox makes impossible to carry out the charge-discharge experiment. Such experiment requires a lot of material in the form of a condensed pill. Here the PANINT are light and their tubular nanostructures can be easily damaged, what is more nanotubes are perpendicularly attached to the thin gold film creating a homogeneous layer. We excluded charge-discharge experiment because it would not reflect the construction of electrode system.
The adsorption/desorption of noble gases is also impossible. We should have c.a. 20 mg of each synthesized materials in an unchanged “form”, as a powder. The specific surface area of pristine CNOs and textural parameters are known. Additionally, conducting polymers have always worse textural parameters in comparison to carbon nanomaterials. We underlined in the manuscript, that the addition of carbon nanomaterials to the polymer matrix increases the porosity of the polymeric material, because this is a commonly observed tendency. However, the textural properties of composites are certainly worse than the parameters of pristine CNOs.

Reviewer 2 Report
This is a very well written article. The introduction is exceptionally clear. Unfortunately, I do not see any of the figures or schemes, which makes it impossible for me to properly review the technical content.
Minor comments on the text:
- Line 73 - "PANInt are also on slightly soluble" should be "PANInt are also only slightly soluble"
- Line 98 - "Their high reactivity when compared with CNs" - Why do CNOs have higher reactivity compared with other CNs? It is due to a specific surface functionalization? Please add this detail.
- Line 120 - What is the typical thickness of the gold film?
Author Response
Thank you for your kind reconsideration of this manuscript. Please find the responses to your comments. The main remark regarding the manuscript. We added all the missing signatures to the Figures and Schemes. Additionally, we added two tables that were missing in the final version of the manuscript (please find attached file). Please find below our answers to all your comments.
We do hope that you will find the article suitable for publication in its present.
Many thanks indeed for your kind help and consideration.
Sincerely yours,
Marta E. Plonska-Brzezinska
Responses to Reviewer #2 comment
Reviewer #2: Line 73 - "PANInt are also on slightly soluble" should be "PANInt are also only slightly soluble"
Comment: It was corrected.
Reviewer #2: Line 98 - "Their high reactivity when compared with CNs" - Why do CNOs have higher reactivity compared with other CNs? It is due to a specific surface functionalization? Please add this detail.
Comment: The high reactivity of CNOs is connected with the spherical structures, and the presence of defects and sp2-hybridized carbon atoms. Some comments are included in the manuscript on Page 3.
Reviewer #2: Line 120 - What is the typical thickness of the gold film?
Comment: The gold film thickness in investigated electrode systems is 100 nm. The information is concluded in subsection 3.3.

Round 2
Reviewer 1 Report
The authors have addressed most of the comment and suggestions raised by the reviewer, thus, the reviewer recommends its publication in polymers.
Reviewer 2 Report
Thank you for the revisions. I recommend publication of this manuscript.